# A Weighting Method Based on the Improved Hesitation of Pythagorean Fuzzy Sets

Xiuli Du [1,2,*], Kun Lu [1,2], Rui Zhou [1,2], Yana Lv [1,2] and Shaoming Qiu [1,2]

1 Communication and Network Laboratory, Dalian University, Dalian 116622, China; lukun@s.dlu.edu.cn (K.L.); zhourui@s.dlu.edu.cn (R.Z.); lvyana@dlu.edu.cn (Y.L.); qiushaoming@dlu.edu.cn (S.Q.)
2 School of Information Engineering, Dalian University, Dalian 116622, China
* Correspondence: duxiuli@dlu.edu.cn

**Abstract:** The existing expert weight determination method for multi-attribute decision making based on the Pythagorean fuzzy number approach does not make sufficient use of the hesitation involved with the decision information, which may cause biased weight assignment. Therefore, to address the issue of unknown expert weights and attribute evaluation based on Pythagorean fuzzy numbers in multi-attribute group decision-making problems, a weight determination method is proposed that improves the treatment of hesitation in Pythagorean fuzzy sets. Firstly, the proximity of experts and similarity of the modified ones are determined according to the evaluation matrix. Then, the expert weights are integrated from the aspects of proximity and corrected similarity to obtain an assembled comprehensive evaluation matrix. Finally, the alternatives are ranked using the PF-TOPSIS method. The results of expert weight analysis and data verification demonstrate that the proposed method fully utilizes expert decision-making information, leading to a significant improvement in the rationality and accuracy of multi-attribute group decision-making problems.

**Keywords:** group decision-making; Pythagorean fuzzy number; Pythagorean fuzzy set; TOPSIS method; objective weight of experts





## 1. Introduction

Multi-criteria group decision-making (MCGDM) is a branch of operations research that is widely used in decision-making processes to determine the optimal solution by evaluating alternative options across multiple conflicting criteria [1–3].

In MCGDM, the weights assigned to experts play a crucial role, particularly during the data aggregation stage. Different expert weights can result in varying evaluation outcomes from the same individual decision matrix. Subjective determination of expert weights can compromise the rigor and scientificity of decision-making. To address this, various methods have been developed for objectively determining expert weights. These methods can be classified into two categories: methods for determining the weights of individual experts and group experts, and methods for determining the weights of individual experts in relation to other experts.

Expert weights are crucial in MCGDM, particularly during the data aggregation stage, as different weights assigned to experts can lead to varied evaluation results from the same individual decision matrix. Subjectively determining expert weights in the MCGDM process undermines the rigor and scientificity of decision-making. Consequently, several methods have been developed to objectively determine expert weights. These methods can be classified into two categories: methods for determining weights of individual experts and group experts, and methods for determining the weights of individual experts in relation to other experts.

(1) Method for calculating individual expert and group expert weights: This category involves fusing the decision-making information provided by each expert into an average

decision-making matrix, known as the ideal decision-making matrix. Subsequently, the similarity between each expert's decision-making matrix and the ideal matrix is used to evaluate and determine the weight of each expert. For example, Yue [4] uses the idea of technique for order preference by similarity to an ideal solution (TOPSIS) to determine the weight, which first establishes an ideal group decision matrix, that is, the mean of the group decision matrix, and secondly determines the expert weight by the similarity between the individual decision matrix and the ideal decision matrix. Zhang and Xu [5] established an optimal model based on consensus maximization to determine the weights of experts by measuring the degree of consensus between individuals and groups. Yue [6] proposed a new group decision matrix method for normalized projection measures that establishes the relationship between each alternative decision and its ideal decision on this basis and then determines the expert weight. Tsao [7] determined the respective projections of the evaluation values of each scheme on the positive ideal and negative ideal solutions in the ambiguous environment of interval intuition, proposed the projection-based comparison index and comprehensive comparison index as the benchmark values for comparison, and increased the percentage value of the comprehensive impact to obtain the ranking results. Li [8] calculated the expert weight by establishing a gray correlation model between each expert weight and the average expert weight. Du [9] proposed an interval number matrix weighted bidirectional projection formula to better measure the similarity relationship between matrices and then proposed an expert weight determination method based on weighted bidirectional projection.

(2) Method for calculating the weights of individual experts and other experts: This category directly assesses and determines the weights by considering the proximity or similarity between each individual expert and other experts. Meng [10] uses the size of the distance to calculate the expert weight, first finding the distance between each expert and the other experts; the larger the distance, the smaller the expert weight, and on the contrary, if the distance is smaller, the expert weight is larger. Wan [11] calculated the weight of experts by the similarity method, first calculating the similarity between each expert and other experts; the greater the similarity, the greater the weight, and the smaller the similarity, the smaller the weight. Meng [12] proposed a one-way projection method between matrices and calculated the similarity from the projection values of experts and other experts, then finally calculated the weight through the similarity. Pang [13] calculated the weights by using the sticking progress of the expert decision-making matrix and the consistency of the ranking scheme. Lin [14] added hesitation information to calculate the weight of experts on the basis of determining the similarity between experts and other experts.

In the two aforementioned types of weight calculation methods, the relationship between individual experts and group experts or between individual experts and other experts is considered, leading to certain limitations in the calculation of expert weights. To address these limitations, this paper proposes a comprehensive approach that integrates both methods and takes into account the relationship among experts. By doing so, it avoids assigning extreme weights to individual experts and adjusts the importance ratio of the two methods based on specific requirements. This approach enhances the scientificity and rationality of the subsequent ranking results.

Yager [15,16] proposed the Pythagorean fuzzy set (PFS) in 2013, which limits the sum of the membership and non-membership squares to be less than or equal to one, enabling the generalization of the intuitionistic fuzzy numbers and providing a new method for processing uncertain information. In recent years, research on Pythagorean fuzzy sets has gained significant attention in various fields such as decision-making, medical diagnosis, and pattern recognition [17].

In 2014, Zhang [18] proposed a new TOPSIS decision-making method based on the work of Yager [15] and effectively solved the multi-objective decision-making problem with the Pythagorean fuzzy information. Zhang [19] proposed a decision-making method based on similarity measurements in the Pythagorean fuzzy environment. In 2015, Peng [20] introduced the division and subtraction operations of the Pythagorean fuzzy number (PFN)

approach and proposed a multi-criterion group decision method by defining the similarity. Gou [21] proposed the PFN fuzzy function and studied its basic properties, such as the continuity, derivability, and differentiability. Peng [22,23] defined the distance formula for PFSs and then derived a similarity calculation method based on the distance formula. These works laid the theoretical foundation for further expanding the widespread application of Pythagorean fuzzy sets [22,24].

By analyzing the relevant literature [4,9–12,14,23,24], we have identified certain limitations in the existing research on weight determination based on similarity. One specific issue is the inadequate consideration of the variations in uncertainties (hesitation) among individual experts' evaluations when calculating expert weights using the ideal solution. This limitation can result in biased weight assignments. To address this concern, we propose an improved approach that incorporates the concept of hesitation correction into the weight determination process. We suggest considering the degree of support, degree of opposition, and degree of hesitation (or uncertainty) when evaluating a scheme's satisfaction of a particular attribute [15,16]. This approach allows for a more comprehensive representation of experts' evaluations and takes into account their uncertainties. In our investigation, we observed situations where two experts evaluated a scheme with different degrees of hesitation but the same degree of membership. Despite the significant difference in hesitations, the weights assigned to the experts using the existing method did not adequately reflect this variation. This discrepancy indicates that the obtained weights may not accurately reflect reality, thereby affecting the credibility of the evaluation results [4,9–12,14,15,18].

The novel contributions of this paper are summarized as follows:

- Proposal of a method to determine weights in the MCGDM problem using Pythagorean fuzzy sets and the Pythagorean fuzzy number approach.
- Incorporation of hesitation into the evaluation of attributes, considering the influence of uncertainty on weights.
- Consideration of the proximity of experts and utilization of a modified similarity measure based on the evaluated matrix.
- Integration of expert weights using proximity and corrected similarity to obtain an assembled comprehensive evaluation matrix.
- Application of the PF-TOPSIS method for ranking the alternatives [23].

These contributions demonstrate the rationality and accuracy of the proposed method.

## 2. Knowledge of Pythagorean Fuzzy Sets

### 2.1. Fundamentals of Pythagorean Fuzzy Sets

**Definition 1** ([15,16]). *Let X be a nonempty set. The PFS of X is denoted by*

$$\xi = \left\{ < x, \xi(u_\xi(x), \lambda_\xi(x)) > | \; x \in X \right\} \tag{1}$$

*where two maps, $u_\xi : X \to [0,1]$ and $\lambda_\xi : X \to [0,1]$, are defined as membership functions and non-membership functions on X, respectively. For each $x \in X$, it is considered and satisfied that $0 \leq u_\xi^2(x) + \lambda_\xi^2(x) \leq 1$. Moreover, $\pi_\xi(x)$ represents x's hesitation with respect to $\xi$, where $\pi_\xi(x) = \sqrt{1 - u_\xi^2(x) - \lambda_\xi^2(x)}$.*

**Definition 2** ([19]). *Given a PFS of $\xi_i = (u_{\xi_i}, \lambda_{\xi_i})(i = 1,2,3,\ldots,n)$, where each $x_i$ has a corresponding weight value $\delta = (\delta_1 \delta_2 \cdots \delta_n)$, satisfying $\sum_{i=1}^n \delta_i = 1$, we can define the Pythagorean fuzzy weighted average (PFWA) operator and the Pythagorean fuzzy weighted geometry (PFWG) operator as Equations (2) and (3), respectively:*

$$
\begin{aligned}
PFWA_\delta(\xi_1, \xi_2, \ldots, \xi_n) &= \delta_1 \xi_1 \oplus \delta_2 \xi_2 \oplus \cdots \oplus \delta_n \xi_n \\
&= \left( \sqrt{1 - \prod_{i=1}^n \left(1 - \left(u_{\xi_1}\right)^2\right)^{\delta_i}}, \prod_{i=1}^n \left(\lambda_{\xi_i}\right)^{\delta_i} \right),
\end{aligned}
\tag{2}
$$

$$PFWG_\delta(\xi_1, \xi_2, \ldots, \xi_n) = \delta_1\xi_1 \otimes \delta_2\xi_2 \otimes \cdots \otimes \delta_n\xi_n$$
$$= \left( \prod_{i=1}^{n} \left(u_{\xi_1}\right)^{\delta_i}, \sqrt{1 - \prod_{i=1}^{n}\left(1 - \left(\lambda_{\xi_i}\right)^2\right)^{\delta_i}} \right). \tag{3}$$

**Definition 3** ([23]). *Given any two PFNs $\xi_1 = (u_{\xi_1}, \lambda_{\xi_1})$ and $\xi_2 = (u_{\xi_2}, \lambda_{\xi_2})$, the normalized Euclidean distance between $\xi_1$ and $\xi_2$ can be defined as follows:*

$$d(\xi_1, \xi_2) = \sqrt{\left(u_{\xi_1}^2 - u_{\xi_2}^2\right)^2 + \left(\lambda_{\xi_1}^2 - \lambda_{\xi_2}^2\right)^2 + \left(\pi_{\xi_1}^2 - \pi_{\xi_2}^2\right)^2}. \tag{4}$$

*2.2. Describe the Question and Problem of the Previous Method*

By analyzing the specialized literature [4,9–12,14,23], we found some shortcomings in the existing theoretical research on determining weights based on similarity. For example, in the calculation of expert weights using the ideal solution, the influence of the differences in uncertainty (hesitation) of the evaluation information from individual experts is not fully considered. This, in turn, leads to biased weight assignment [23]. For a given scheme $A_i$ and attribute $G_j$, the intuitionistic fuzzy number $a = (u_a, \lambda_a)$ is used. $u_a$ represents the degree to which scheme $A_i$ satisfies attribute $G_j$ (degree of support), $\lambda_a$ represents the degree to which scheme $A_i$ does not satisfy attribute $G_j$ (degree of opposition), and $\pi_A(x)$ indicates the degree to which scheme $A_i$ exhibits hesitation or uncertainty with respect to attribute $G_j$ (degree of hesitation, degree of uncertainty) [15,16]. Suppose that ten people vote for a certain proposal as follows: five people in favor, three people against, and two abstentions. Hence, the degree of support $u_a = 0.5$, the degree of opposition $\lambda_a = 0.3$, and the degree of hesitation $\pi_a(x) = 0.2$ [18]. When the degree of membership is the same but the hesitation differs significantly, for instance, when a scheme is evaluated by two experts, $\varepsilon_1$ and $\varepsilon_2$, the following values are given: $\xi_1 = (u_1, \lambda_1) = (0.01, 0.4358)$, $\pi_1 = 0.9$, $\xi_2 = (u_2, \lambda_2) = (0.01, 0.979)$, and $\pi_2 = 0.2$. The ideal solution (mean) for the evaluation information is $\bar{\xi} = (0.01, 0.7658)$, $\bar{\pi} = 0.643$. The distance between the evaluation information and the ideal solution is $d_1 = 0.561, d_2 = 0.527$ [10,23]. The weight assignment method used is based on both the individual expert and group expert calculation methods. The weights assigned to the two experts, $\varepsilon_1, \varepsilon_2$, are $w_1 = 0.509$, $w_2 = 0.492$, respectively. It can be observed that 0.509 is greater than 0.492. As the hesitation itself represents the uncertainty of an expert, it is evident that the hesitations of the two experts are significantly different. The degree of hesitation indicates the magnitude of uncertainty within an expert's evaluation [5]. However, despite the notable difference in hesitation, the obtained weights for the two experts are not significantly different. This discrepancy suggests that the weight obtained through this method does not accurately reflect reality, thereby impacting the credibility of the evaluation results [4,9–12,14,15,18].

**3. Method for Determination of Expert Weights Considering Hesitation**

For the MCGDM problem containing PFN, there are $l$ alternatives, $\mathcal{A}_1, \mathcal{A}_2, \cdots, \mathcal{A}_l$, each of which has $m$ attributes $C_1, C_2, \cdots, C_m$ with $\mathcal{W}_j$ as the weight of attribute $C_j$, and $\sum_{j=1}^{m} \mathcal{W}_j = 1$. Let $n$ experts $\varepsilon = \{\varepsilon_1, \varepsilon_2, \ldots, \varepsilon_n\}$ participate in decision-making; $\delta_i$ is the weight of expert $\varepsilon_i$, and $\sum_{i=1}^{n} \delta_i = 1$. Let expert $\varepsilon_k(k = 1, 2, 3, \ldots, n)$ is the evaluation matrix for attribute $C_j(j = 1, 2, 3, \ldots, m)$ under scheme $\mathcal{A}_i(i = 1, 2, 3, \ldots, l)$ with $\mathcal{X}^{(k)} = (x_{ij}^{(k)})_{l*m}$, where $x_{ij}^{(k)} = \left(u_{ij}^{(k)}, \lambda_{ij}^{(k)}, \pi_{ij}^{(k)}\right)$ is the property of expert $\varepsilon_k$ for scenario $\mathcal{A}_i$ with respect to

attribute $C_j$. Therefore, the MCGDM problem can be expressed in the form of the following matrix [9]:

$$\mathcal{X}^{(k)} = \begin{matrix} & \begin{matrix} C_1 & C_2 & \cdots & C_m \end{matrix} \\ \begin{matrix} \mathcal{A}_1 \\ \mathcal{A}_2 \\ \vdots \\ \mathcal{A}_l \end{matrix} & \begin{bmatrix} \left(u_{11}^{(k)}, \lambda_{11}^{(k)}, \pi_{11}^{(k)}\right) & \left(u_{12}^{(k)}, \lambda_{12}^{(k)}, \pi_{12}^{(k)}\right) & \cdots & \left(u_{1m}^{(k)}, \lambda_{1m}^{(k)}, \pi_{1m}^{(k)}\right) \\ \left(u_{21}^{(k)}, \lambda_{21}^{(k)}, \pi_{21}^{(k)}\right) & \left(u_{22}^{(k)}, \lambda_{22}^{(k)}, \pi_{22}^{(k)}\right) & \cdots & \left(u_{2m}^{(k)}, \lambda_{2m}^{(k)}, \pi_{2m}^{(k)}\right) \\ \vdots & \vdots & \ddots & \vdots \\ (u_{l1}^{(k)}, \lambda_{l1}^{(k)}, \pi_{l1}^{(k)}) & \left(u_{l2}^{(k)}, \lambda_{l2}^{(k)}, \pi_{l2}^{(k)}\right) & \cdots & \left(u_{lm}^{(k)}, \lambda_{lm}^{(k)}, \pi_{lm}^{(k)}\right) \end{bmatrix} \end{matrix}.$$

The method for determining weights in the decision process is explained stepwise below:

Step 1: Establish a decision matrix, $\mathcal{X}^{(k)} = (x_{ij}^{(k)})_{l*n} = \left(u_{ij}^{(k)}, \lambda_{ij}^{(k)}, \pi_{ij}^{(k)}\right)_{l*m}$, for each expert.

Step 2: Determine the average decision matrix $\overline{\mathcal{X}}$ (ideal matrix). In the Pythagorean fuzzy decision matrix (PFDM) process, the personal decision matrices of all the experts are aggregated into an ideal matrix. In this aggregation, the ratings provided by the experts are evaluated based on their proximity to the average rating. If an expert's rating is closer to the average, it is considered a better rating, whereas a rating further away from the average is considered worse. By aggregating the individual decision matrices into an ideal matrix, a comprehensive evaluation is obtained for the decision-making process [11,23].

$\overline{\mathcal{X}} = (\overline{x}_{ij})_{l*n}$ is calculated using Equation (2), where expert $\varepsilon_k$ is weighted $\delta_k (k = 1, 2, 3, \ldots, n)$, and $\delta_1 = \delta_2 = \cdots = \delta_k = \frac{1}{n}$.

Step 3: Determine the similarity matrix [14]. The distance $d_{ij}^{(k)}\left(x_{ij}^{(k)}, \overline{x}_{ij}\right)$ between expert $\varepsilon_k$ and $\overline{\mathcal{X}}$ is calculated by using Equation (4), and then all the distances are aggregated into a distance matrix $D^{(k)} = (d_{ij}^{(k)})_{l*n}$. Finally, the similarity matrix $S^{(k)} = (s^{(k)})_{l*n}$ of each expert and $\overline{\mathcal{X}}$ are obtained by using Equation (5).

$$s_{ij}^{(k)} = \sqrt{2} - d_{ij}^{(k)}\left(x_{ij}^{(k)}, \overline{x}_{ij}\right) = \sqrt{2} - \sqrt{\left(\left(u_{ij}^{(k)}\right)^2 - (\overline{\mu}_{ij})^2\right)^2 + \left(\left(\lambda_{ij}^{(k)}\right)^2 - (\overline{\lambda}_{ij})^2\right)^2 + \left(\left(\pi_{ij}^{(k)}\right)^2 - (\overline{\pi}_{ij})^2\right)^2}. \quad (5)$$

Step 4: Fix the similarity matrix. After obtaining the hesitation information about expert $\varepsilon_k$ as stated in step 1 and similarity matrix as stated in step 3, the similarity is corrected by using Equation (6), and the hesitation correction coefficient $\theta$ of expert $\varepsilon_k$ to $\mathcal{A}_i$ under attribute $C_j$ is recorded.

$$\theta_{ij}^{(k)} = \cos \frac{\Pi(\pi_{ij}^{(k)})^2}{2}, (0 \leq \pi_{ij}^{(k)} \leq 1) \quad (6)$$

where $\pi_{ij}^{(k)}$ is the hesitation of the current expert $\varepsilon_k$ to $\mathcal{A}_i$ in attribute $C_j$, and $\Pi = 3.14159\ldots$.

This function is chosen in this article as it satisfies the requirements of the following concepts considered:

(1) The selected hesitation value satisfies the given conditions.
(2) If the degree of hesitation is 0, no correction is needed (the weight determined by the similarity degree remains unchanged), and when the hesitation value is 1, the weight correction is 0 (the weight determined by the similarity degree is 0).
(3) The effect of correction becomes more pronounced as the hesitation increases, resulting in a steeper slope.

The corrected similarity $s*_{ij}^{(k)}$ can be expressed by Equation (7).

$$s*_{ij}^{(k)} = \theta_{ij}^{(k)} \times s_{ij}^{(k)} = \cos \frac{\Pi(\pi_{ij}^{(k)})^2}{2} \times s_{ij}^{(k)} \quad (7)$$

Step 5: Determine the proximity matrix of each individual to other individuals. In group decision-making, the number of selected experts is typically limited to ten or fewer. If an evaluated individual's assessment is unreasonable, it can have a significant impact on the group decision-making process. Therefore, when calculating the weight, it is important to consider the proximity between the individual in question and other individuals, which reflects the similarity of attribute evaluation information among experts. The greater the proximity, the higher the weight that should be assigned to the expert [25,26].

Relative to expert $\varepsilon_k$, the mean value of the evaluated information of the remaining experts after excluding expert $\varepsilon_k$ is $\widetilde{x}_{ij}^{(k)}$, and the distance between expert $\varepsilon_k$ and the rest of the expert evaluation information is $d\left(e_{ij}^{(k)}, \widetilde{e}_{ij}^k\right)$. The proximity $p_{ij}^{(k)}$, which can be calculated using Equation (4), is expressed by Equation (8) as follows:

$$p_{ij}^{(k)} = \sqrt{2} - d\left(e_{ij}^{(k)}, \widetilde{e}_{ij}^k\right) \tag{8}$$

The larger the value of $p_{ij}^{(k)}$, the closer the information given by expert $\varepsilon_k$ to the information given by all other experts. Therefore, the weight of expert $\varepsilon_k$ under attribute $C_j$ should also be greater.

Step 6: Determine the weight of experts [25]. Combining the proximity given by Equation (8) and similarity given by Equation (7), we control the combined weight $\overline{\delta}_j^{(k)}$ of expert $\varepsilon_k$ under attribute $C_j$ with the help of a parameter, i.e,

$$\overline{\delta}_j^{(k)} = \eta s_{ij}^{*(k)} + (1 - \eta) p_{ij}^{(k)} \tag{9}$$

By changing the value of parameter $\eta$ in Equation (9), the combined weights of experts can be determined evenly according to the similarity and proximity. In particular, when $\eta = 0$, $\delta_j^{(k)}$ depends only on proximity $p_{ij}^{(k)}$. When $\eta = 1$, only similarity $s_{ij}^{*(k)}$ is dependent. In order to comprehensively consider the proximity and similarity of experts, $\eta = 0.5$ is preferred.

Normalization of the combined weight $\overline{\delta}_j^{(k)}$ yields the weight of the expert under attribute $C_j$ as expressed by Equation (10).

$$\delta_j^{(k)} = \overline{\delta}_j^{(k)} / \sum_{k=1}^{n} \overline{\delta}_j^{(k)} (j \in m, k \in n) \tag{10}$$

## 4. Group Decision-Making Methods

To effectively address the aforementioned MCGDM problem with PFN, we employ the TOPSIS method to facilitate group decision-making.

The decision-making process of the PF-TOPSIS method considering hesitation is as follows [23]:

Step 1: Determine the aggregate PFDM $\mathcal{X} = (x_{ij})_{l*n}$. In the decision-making process, the individual opinions of all experts need to be combined into one collective opinion. Combine Equation (10) $\delta_j^{(k)}$ with Equation (2) to calculate $\mathcal{X} = (x_{ij})_{l*n}$, where $x_{ij} = \left(u_{\mathcal{A}_i}(C_j), \lambda_{\mathcal{A}_i}(C_j), \pi_{\mathcal{A}_i}(C_j)\right), i = 1, 2, \ldots, l, j = 1, 2, \ldots, m$.

$$\mathcal{X} = \begin{array}{c} \mathcal{A}_1 \\ \mathcal{A}_2 \\ \vdots \\ \mathcal{A}_l \end{array} \begin{bmatrix} \begin{array}{cccc} C_1 & C_2 & \cdots & C_m \end{array} \\ \left(u_{\mathcal{A}_1}(C_1), \lambda_{\mathcal{A}_1}(C_1), \pi_{\mathcal{A}_1}(C_1)\right) & \left(u_{\mathcal{A}_1}(C_2), \lambda_{\mathcal{A}_1}(C_2), \pi_{\mathcal{A}_1}(C_2)\right) & \cdots & \left(u_{\mathcal{A}_1}(C_m), \lambda_{\mathcal{A}_1}(C_m), \pi_{\mathcal{A}_1}(C_m)\right) \\ \left(u_{\mathcal{A}_2}(C_1), \lambda_{\mathcal{A}_2}(C_1), \pi_{\mathcal{A}_2}(C_1)\right) & \left(u_{\mathcal{A}_2}(C_2), \lambda_{\mathcal{A}_2}(C_2), \pi_{\mathcal{A}_2}(C_2)\right) & \cdots & \left(u_{\mathcal{A}_2}(C_m), \lambda_{\mathcal{A}_2}(C_m), \pi_{\mathcal{A}_2}(C_m)\right) \\ \vdots & \vdots & \ddots & \vdots \\ \left(u_{\mathcal{A}_l}(C_1), \lambda_{\mathcal{A}_l}(C_1), \pi_{\mathcal{A}_l}(C_1)\right) & \left(u_{\mathcal{A}_l}(C_2), \lambda_{\mathcal{A}_l}(C_2), \pi_{\mathcal{A}_l}(C_2)\right) & \cdots & \left(u_{\mathcal{A}_l}(C_m), \lambda_{\mathcal{A}_l}(C_m), \pi_{\mathcal{A}_l}(C_m)\right) \end{bmatrix}. \tag{11}$$

Step 2: Determine the aggregation-weighting PFDM $\mathcal{X}' = (x'_{ij})_{l \times m}$. Combining Equation (11) $\mathcal{X} = (x_{ij})_{l*n}$ with the attribute weight matrix $\mathcal{W}$ and then using the multiplication operator [18] Equation (12) yields $\mathcal{X}' = (x'_{ij})_{l \times m}$, where $x'_{ij} = (u_{\mathcal{A}_i w}(C_j), \lambda_{\mathcal{A}_i w}(C_j), \pi_{\mathcal{A}_i w}(C_j))$.

$$x'_{ij} = x_{ij} \otimes w_j$$
$$= \left( u_{\mathcal{A}_i}(C_j) \cdot u_{\mathcal{W}}(C_j), \sqrt{\lambda^2_{\mathcal{A}_i}(C_j) + \lambda^2_{\mathcal{W}}(C_j) - \lambda^2_{\mathcal{A}_i}(C_j) \cdot \lambda^2_{\mathcal{W}}(C_j)} \right) \quad (12)$$
$$\pi_{\mathcal{A}_i \mathcal{W}}(C_j) = \sqrt{1 - u_{\mathcal{A}_i}(C_j) \cdot u_{\mathcal{W}}(C_j) - \lambda^2_{\mathcal{A}_i}(C_j) - \lambda^2_{\mathcal{W}}(C_j) + \lambda^2_{\mathcal{A}_i}(C_j) \cdot \lambda^2_{\mathcal{W}}(C_j)}.$$

Aggregation-weighted PFDM $\mathcal{X}' = (x'_{ij})_{l \times m}$:

$$\mathcal{X}' = \begin{array}{c} \\ \mathcal{A}_1 \\ \mathcal{A}_2 \\ \vdots \\ \mathcal{A}_l \end{array} \begin{bmatrix} \overset{C_1}{\begin{pmatrix} u_{\mathcal{A}_1 W}(C_1), \lambda_{\mathcal{A}_1 W}(C_1) \\ , \pi_{\mathcal{A}_1 W}(C_1) \end{pmatrix}} & \overset{C_2}{\begin{pmatrix} u_{\mathcal{A}_1 W}(C_2), \lambda_{\mathcal{A}_1 W}(C_2) \\ , \pi_{\mathcal{A}_1 W}(C_2) \end{pmatrix}} & \cdots & \overset{C_m}{\begin{pmatrix} u_{\mathcal{A}_1 W}(C_m), \lambda_{\mathcal{A}_1 W}(C_m) \\ , \pi_{\mathcal{A}_1 W}(C_m) \end{pmatrix}} \\ \begin{pmatrix} u_{\mathcal{A}_2 W}(C_1), \lambda_{\mathcal{A}_2 W}(C_1) \\ , \pi_{\mathcal{A}_2 W}(C_1) \end{pmatrix} & \begin{pmatrix} u_{\mathcal{A}_2 W}(C_2), \lambda_{\mathcal{A}_2 W}(C_2) \\ , \pi_{\mathcal{A}_2 W}(C_2) \end{pmatrix} & \cdots & \begin{pmatrix} u_{\mathcal{A}_2 W}(C_m), \lambda_{\mathcal{A}_2 W}(C_m) \\ , \pi_{\mathcal{A}_2 W}(C_m) \end{pmatrix} \\ \vdots & \vdots & \ddots & \vdots \\ \begin{pmatrix} u_{\mathcal{A}_l W}(C_1), \lambda_{\mathcal{A}_l W}(C_1) \\ , \pi_{\mathcal{A}_l W}(C_1) \end{pmatrix} & \begin{pmatrix} u_{\mathcal{A}_l W}(C_2), \lambda_{\mathcal{A}_l W}(C_2) \\ , \pi_{\mathcal{A}_l W}(C_2) \end{pmatrix} & \cdots & \begin{pmatrix} u_{\mathcal{A}_l W}(C_m), \lambda_{\mathcal{A}_l W}(C_m) \\ , \pi_{\mathcal{A}_l W}(C_m) \end{pmatrix} \end{bmatrix}. \quad (13)$$

Step 3: Determine the positive and negative ideal solutions. Let $J_1$ and $J_2$ be revenue-type attributes and cost-type attributes, respectively. The Pythagorean fuzzy positive ideal solution (PFPIS) $\mathcal{A}^+$ and Pythagorean fuzzy negative ideal solution (PFNIS) $\mathcal{A}^-$ are defined as:

$$\mathcal{A}^+ = \{ \langle C_j, u_{\mathcal{A}^+ w}, \lambda_{\mathcal{A}^+ w} \rangle \mid C_j \in C, j = 1, 2, \ldots, m \}, \quad (14)$$

$$\mathcal{A}^- = \{ \langle C_j, u_{\mathcal{A}^- W}, \lambda_{\mathcal{A}^- W} \rangle \mid C_j \in C, j = 1, 2, \ldots, m \}, \quad (15)$$

moreover,

$$\mu_{\mathcal{A}^+ \mathcal{W}}(C_j) = \begin{cases} \max\limits_{1 \leq i \leq l} u_{\mathcal{A}_i \mathcal{W}}(C_j) & \text{if } C_j \in J_1 \\ \min\limits_{1 \leq i \leq l} u_{\mathcal{A}_i \mathcal{W}}(C_j) & \text{if } C_j \in J_2 \end{cases}, \lambda_{\mathcal{A}^+ \mathcal{W}}(C_j) = \begin{cases} \min\limits_{1 \leq i \leq l} \lambda_{\mathcal{A}_i \mathcal{W}}(C_j) & \text{if } C_j \in J_1, \\ \max\limits_{1 \leq i \leq l} \lambda_{\mathcal{A}_i \mathcal{W}}(C_j) & \text{if } C_j \in J_2, \end{cases} \quad (16)$$

$$\mu_{\mathcal{A}^- \mathcal{W}}(C_j) = \begin{cases} \min\limits_{1 \leq i \leq l} u_{\mathcal{A}_i \mathcal{W}}(C_j) & \text{if } C_j \in J_1 \\ \max\limits_{1 \leq i \leq l} u_{\mathcal{A}_i \mathcal{W}}(C_j) & \text{if } C_j \in J_2 \end{cases}, \lambda_{\mathcal{A}^- \mathcal{W}}(C_j) = \begin{cases} \max\limits_{1 \leq i \leq l} \lambda_{\mathcal{A}_i \mathcal{W}}(C_j) & \text{if } C_j \in J_1, \\ \min\limits_{1 \leq i \leq l} \lambda_{\mathcal{A}_i \mathcal{W}}(C_j) & \text{if } C_j \in J_2. \end{cases} \quad (17)$$

Step 4: Determine the distance between the positive and negative ideal solutions and the alternatives. PFPIS $\mathcal{A}^+$ is usually absent in real-life decision problems, where $\mathcal{A}^+ \notin \mathcal{A}$; otherwise, it would be the ultimate suitable choice for MCGDM problems. Conversely, PFNIS $\mathcal{A}^-$ is the worst choice to solve the MCGDM problem, where $\mathcal{A}^- \notin \mathcal{A}$. Therefore, we continue to determine the distance of each alternative from PFPIS and PFNIS by defining a distance metric. To achieve this, the normalized Euclidean distance between PFSs is defined [22] as follows:

$$D(\mathcal{A}_i, \mathcal{A}^+) = \sqrt{\frac{1}{2m} \sum_{j=1}^{m} \left[ \begin{array}{l} \left( u^2_{\mathcal{A}_i w}(C_j) - \mu^2_{\mathcal{A}^+ w}(C_j) \right)^2 + \left( \lambda^2_{\mathcal{A}_i w}(C_j) - \lambda^2_{\mathcal{A}^+ w}(C_j) \right)^2 \\ + \left( \pi^2_{\mathcal{A}_i w}(C_j) - \pi^2_{\mathcal{A}^+ w}(C_j) \right)^2 \end{array} \right]} \quad (18)$$

$$D(\mathcal{A}_i, \mathcal{A}^-) = \sqrt{\frac{1}{2m}\sum_{j=1}^{m}\left[\begin{array}{c}\left(u^2_{\mathcal{A}_i w}(C_j) - u^2_{\mathcal{A}^- w}(C_j)\right)^2 + \left(\lambda^2_{\mathcal{A}_i w}(C_j) - \lambda^2_{\mathcal{A}^- w}(C_j)\right)^2 \\ + \left(\pi^2_{\mathcal{A}_i w}(C_j) - \pi^2_{\mathcal{A}^- w}(C_j)\right)^2\end{array}\right]} \tag{19}$$

$$i = 1, 2, 3, \ldots, l.$$

Step 5: Determine the composite evaluation index of the alternatives. Obviously, the closer option $\mathcal{A}_i$ is to $\mathcal{A}^+$ and the farther away it is from $\mathcal{A}^-$, the greater the value of its composite evaluation index $\rho(\mathcal{A}_i)$, and thus option $\mathcal{A}_i$ is a better choice than other options [25,26], where $\rho(\mathcal{A}_i) \leq 0 (i = 1, 2, \ldots, l)$.

$$\rho(\mathcal{A}_i) = \frac{D(\mathcal{A}_i, \mathcal{A}^-)}{D_{\max}(\mathcal{A}_i, \mathcal{A}^-)} - \frac{D(\mathcal{A}_i, \mathcal{A}^+)}{D_{\min}(\mathcal{A}_i, \mathcal{A}^+)}, i = 1, 2, \ldots, l,$$

$$\mathcal{A}^* := \{\mathcal{A}_i : (i : \rho(\mathcal{A}_i) = \max_{1 \leq k \leq l} \rho(\mathcal{A}_k))\}. \tag{20}$$

## 5. Case Study

To verify the rationality and effectiveness of the proposed method for determining expert weights by considering the degree of hesitation in group decision-making, a case study is conducted in two parts: expert weight determination and group decision-making.

### 5.1. Determination of Expert Weights
5.1.1. Case Description

In Section 5.1, we discuss the determination of expert weights, building upon the theoretical foundation established in Section 3, specifically in the section titled 'Method for determination of expert weights considering hesitation'. It is important to note that a comprehensive theoretical description of the expert weight determination has already been provided in Section 3. Therefore, in Section 5.1, we aim to provide a concise overview and highlight the key aspects of the expert weight determination method. This approach allows us to avoid redundancy and ensure a clear and focused discussion. By referring back to the theoretical framework established in Section 3, readers can gain a deeper understanding of the subsequent analysis and results presented in this section.

The National Academy of Science of Pakistan is dedicated to the national youth education and is the largest educational network in Pakistan. In order to meet the growing educational needs in Faisalabad, they decided to establish a university campus there to provide quality education and a good learning environment for students. After a visit to the city and a pre-assessment, they planned to choose one of five alternative sites $\mathcal{A}_i (i = 1, 2, \ldots, 5)$ as the best place to build a university campus. In order to address this decision-making issue, the owners of the educational institutions formed a committee of three experts, including a legal expert, $\varepsilon_1$, an investment expert, $\varepsilon_2$, and a population expert, $\varepsilon_3$, to evaluate the alternative locations based on the following four attributes:

C1: Policy and theoretical perspective;
C2: Convenience and livability of teachers and students;
C3: Construction cost;
C4: Economic development of the region.

5.1.2. Determination of Expert Weights

Weight determination consists of the following steps:
Step 1: Establish the decision matrices $\mathcal{X}^{(1)}, \mathcal{X}^{(2)}, \mathcal{X}^{(3)}$ of legal expert $\varepsilon_1$, investment expert $\varepsilon_2$, and population expert $\varepsilon_3$, respectively, as Tables 1–3:

**Table 1.** Pythagorean Fuzzy Decision Matrix $\mathcal{X}^{(1)}$.

|       | C1                     | C2                     | C3                     | C4                     |
|-------|------------------------|------------------------|------------------------|------------------------|
| $A_1$ | 0.3162, 0.3000, 0.9000 | 0.5000, 0.6000, 0.6200 | 0.7000, 0.3500, 0.6200 | 0.6000, 0.5000, 0.6200 |
| $A_2$ | 0.2753, 0.3100, 0.9100 | 0.2000, 0.8000, 0.5700 | 0.6000, 0.5000, 0.6200 | 0.1000, 0.9000, 0.4200 |
| $A_3$ | 0.3434, 0.3000, 0.8900 | 0.4500, 0.7000, 0.5500 | 0.4500, 0.7000, 0.5500 | 0.4000, 0.7500, 0.5300 |
| $A_4$ | 0.2636, 0.2900, 0.9200 | 0.2000, 0.8000, 0.5700 | 0.4500, 0.7000, 0.5500 | 0.2000, 0.8000, 0.5700 |
| $A_5$ | 0.2742, 0.2800, 0.9200 | 0.8000, 0.2500, 0.5500 | 0.9000, 0.2000, 0.3900 | 0.1000, 0.9000, 0.4200 |

**Table 2.** Pythagorean Fuzzy Decision Matrix $\mathcal{X}^{(2)}$.

|       | C1                     | C2                     | C3                     | C4                     |
|-------|------------------------|------------------------|------------------------|------------------------|
| $A_1$ | 0.7533, 0.6500, 0.1000 | 0.8000, 0.2500, 0.5500 | 0.9000, 0.2000, 0.3900 | 0.4000, 0.7500, 0.5300 |
| $A_2$ | 0.7794, 0.6200, 0.0900 | 0.5000, 0.6000, 0.6200 | 0.4000, 0.7500, 0.5300 | 0.9000, 0.2000, 0.3900 |
| $A_3$ | 0.7893, 0.6100, 0.0700 | 0.1000, 0.9000, 0.4200 | 0.6000, 0.5000, 0.6200 | 0.6000, 0.5000, 0.6200 |
| $A_4$ | 0.7432, 0.6600, 0.1100 | 0.2000, 0.8000, 0.5700 | 0.4000, 0.7500, 0.5300 | 1.0000, 0.0000, 0.0000 |
| $A_5$ | 0.7589, 0.6400, 0.1200 | 0.4000, 0.7500, 0.5300 | 0.4000, 0.7500, 0.5300 | 0.8000, 0.2500, 0.5500 |

**Table 3.** Pythagorean Fuzzy Decision Matrix $\mathcal{X}^{(3)}$.

|       | C1                     | C2                     | C3                     | C4                     |
|-------|------------------------|------------------------|------------------------|------------------------|
| $A_1$ | 0.6196, 0.4400, 0.6500 | 0.2000, 0.8000, 0.5700 | 0.6000, 0.5000, 0.6200 | 0.2000, 0.8000, 0.5700 |
| $A_2$ | 0.6657, 0.4000, 0.6300 | 0.2000, 0.8000, 0.5700 | 0.2000, 0.8000, 0.5700 | 0.4500, 0.7000, 0.5500 |
| $A_3$ | 0.6155, 0.4600, 0.6400 | 0.9000, 0.2000, 0.3900 | 0.9000, 0.2000, 0.3900 | 0.7000, 0.3500, 0.6200 |
| $A_4$ | 0.6427, 0.4500, 0.6200 | 0.2000, 0.8000, 0.5700 | 1.0000, 0.0000, 0.0000 | 0.7000, 0.3500, 0.6200 |
| $A_5$ | 0.5861, 0.4700, 0.6600 | 0.4500, 0.7000, 0.5500 | 0.9000, 0.2000, 0.3900 | 0.8000, 0.2500, 0.5500 |

Step 2: Determine the average decision matrix $\overline{\mathcal{X}} = (\overline{x}_{ij})_{l*n}$ of legal expert $\varepsilon_1$, investment expert $\varepsilon_2$, and population expert $\varepsilon_3$ using Equation (2) as Table 4:

**Table 4.** Mean Aggregation of Pythagorean Fuzzy Decision Matrix $\overline{\mathcal{X}}$.

|       | C1                     | C2                     | C3                     | C4                     |
|-------|------------------------|------------------------|------------------------|------------------------|
| $A_1$ | 0.6154, 0.4411, 0.6533 | 0.6020, 0.4932, 0.6279 | 0.7773, 0.3271, 0.5374 | 0.4448, 0.6694, 0.5950 |
| $A_2$ | 0.6428, 0.4252, 0.6371 | 0.3403, 0.7268, 0.5965 | 0.4448, 0.6694, 0.5950 | 0.6846, 0.5013, 0.5292 |
| $A_3$ | 0.6394, 0.4383, 0.6317 | 0.6846, 0.5013, 0.5292 | 0.7352, 0.4121, 0.5381 | 0.5919, 0.5082, 0.6256 |
| $A_4$ | 0.6121, 0.4416, 0.6559 | 0.2000, 0.8000, 0.5657 | 0.9741, 0.0374, 0.2231 | 0.9767, 0.0304, 0.2125 |
| $A_5$ | 0.6032, 0.4383, 0.6664 | 0.6145, 0.5082, 0.6035 | 0.8296, 0.3107, 0.4640 | 0.7040, 0.3832, 0.5980 |

Step 3: Determine the similarity matrix. Using Equation (4), the distance $d_{ij}^{(k)}\left(x_{ij}^{(k)}, \overline{x}_{ij}\right)$ between expert $\varepsilon_k$ and ideal solution $\overline{\mathcal{X}}$ is calculated and integrated into the distance matrix $D^{(k)} = (d_{ij}^{(k)})_{l*n}$, and then the similarity matrix $S^{(k)} = (s^{(k)})_{l*n}$ of each expert and ideal solution $\overline{\mathcal{X}}$ are obtained by the similarity given by Equation (5). The similarity matrices $S^{(1)}$, $S^{(2)}$, and $S^{(3)}$ of $\varepsilon_1$, $\varepsilon_2$, and $\varepsilon_3$, respectively, are as Tables 5–7:

**Table 5.** Similarity Matrix $S^{(1)}$.

|  | C1 | C2 | C3 | C4 |
|---|---|---|---|---|
| $A_1$ | 0.9290 | 1.2520 | 1.2650 | 1.1560 |
| $A_2$ | 0.8672 | 1.2760 | 1.1560 | 0.6840 |
| $A_3$ | 0.9147 | 1.0560 | 0.9484 | 1.0390 |
| $A_4$ | 0.8864 | 1.4090 | 0.4871 | 0.2645 |
| $A_5$ | 0.9061 | 1.0810 | 1.2660 | 0.5725 |

**Table 6.** Similarity Matrix $S^{(2)}$.

|  | C1 | C2 | C3 | C4 |
|---|---|---|---|---|
| $A_1$ | 0.9030 | 1.0700 | 1.1580 | 1.2730 |
| $A_2$ | 0.9269 | 1.1970 | 1.2730 | 0.9929 |
| $A_3$ | 0.9309 | 0.6840 | 1.1950 | 1.4000 |
| $A_4$ | 0.9001 | 1.4090 | 0.4190 | 1.3500 |
| $A_5$ | 0.8880 | 1.0310 | 0.7068 | 1.2380 |

**Table 7.** Similarity Matrix $S^{(3)}$.

|  | C1 | C2 | C3 | C4 |
|---|---|---|---|---|
| $A_1$ | 1.4070 | 0.8983 | 1.1160 | 1.1640 |
| $A_2$ | 1.3770 | 1.2760 | 1.1640 | 1.0560 |
| $A_3$ | 1.3770 | 0.9929 | 1.0850 | 1.2190 |
| $A_4$ | 1.3540 | 1.4090 | 1.3430 | 0.8268 |
| $A_5$ | 1.3780 | 1.1170 | 1.2660 | 1.2380 |

Step 4: Determine the similarity correction matrix. Using Equation (7) to correct the similarity, the similarity correction matrices $S^{*(1)}$, $S^{*(2)}$, and $S^{*(3)}$ of $\varepsilon_1$, $\varepsilon_2$, and $\varepsilon_3$. In this study, we present the results of our analysis, which are summarized in Tables 8–10. Specifically, Figure 1 corresponds to Table 8, Figure 2 corresponds to Table 9, and Figure 3 corresponds to Table 10:

**Table 8.** Similarity Correction Matrix $S^{*(1)}$.

|  | C1 | C2 | C3 | C4 |
|---|---|---|---|---|
| $A_1$ | 0.2732 | 1.0310 | 1.0410 | 0.9519 |
| $A_2$ | 0.2314 | 1.1130 | 0.9519 | 0.6579 |
| $A_3$ | 0.2935 | 0.9390 | 0.8434 | 0.9392 |
| $A_4$ | 0.2118 | 1.2300 | 0.4331 | 0.2308 |
| $A_5$ | 0.2165 | 0.9613 | 1.2300 | 0.5507 |

**Table 9.** Similarity Correction Matrix $S^{*(2)}$.

|       | C1     | C2     | C3     | C4     |
|-------|--------|--------|--------|--------|
| $A_1$ | 0.9029 | 0.9519 | 1.1250 | 1.1510 |
| $A_2$ | 0.9268 | 0.9854 | 1.1510 | 0.9647 |
| $A_3$ | 0.9308 | 0.6579 | 0.9838 | 1.1520 |
| $A_4$ | 0.9000 | 1.2300 | 0.3788 | 1.3500 |
| $A_5$ | 0.8878 | 0.9323 | 0.6392 | 1.1010 |

**Table 10.** Similarity Correction Matrix $S^{*(3)}$.

|       | C1     | C2     | C3     | C4     |
|-------|--------|--------|--------|--------|
| $A_1$ | 1.1090 | 0.7839 | 0.9183 | 1.0160 |
| $A_2$ | 1.1180 | 1.1130 | 1.0160 | 0.9390 |
| $A_3$ | 1.1020 | 0.9647 | 1.0540 | 1.0040 |
| $A_4$ | 1.1150 | 1.2300 | 1.3430 | 0.6806 |
| $A_5$ | 1.0680 | 0.9935 | 1.2300 | 1.1010 |

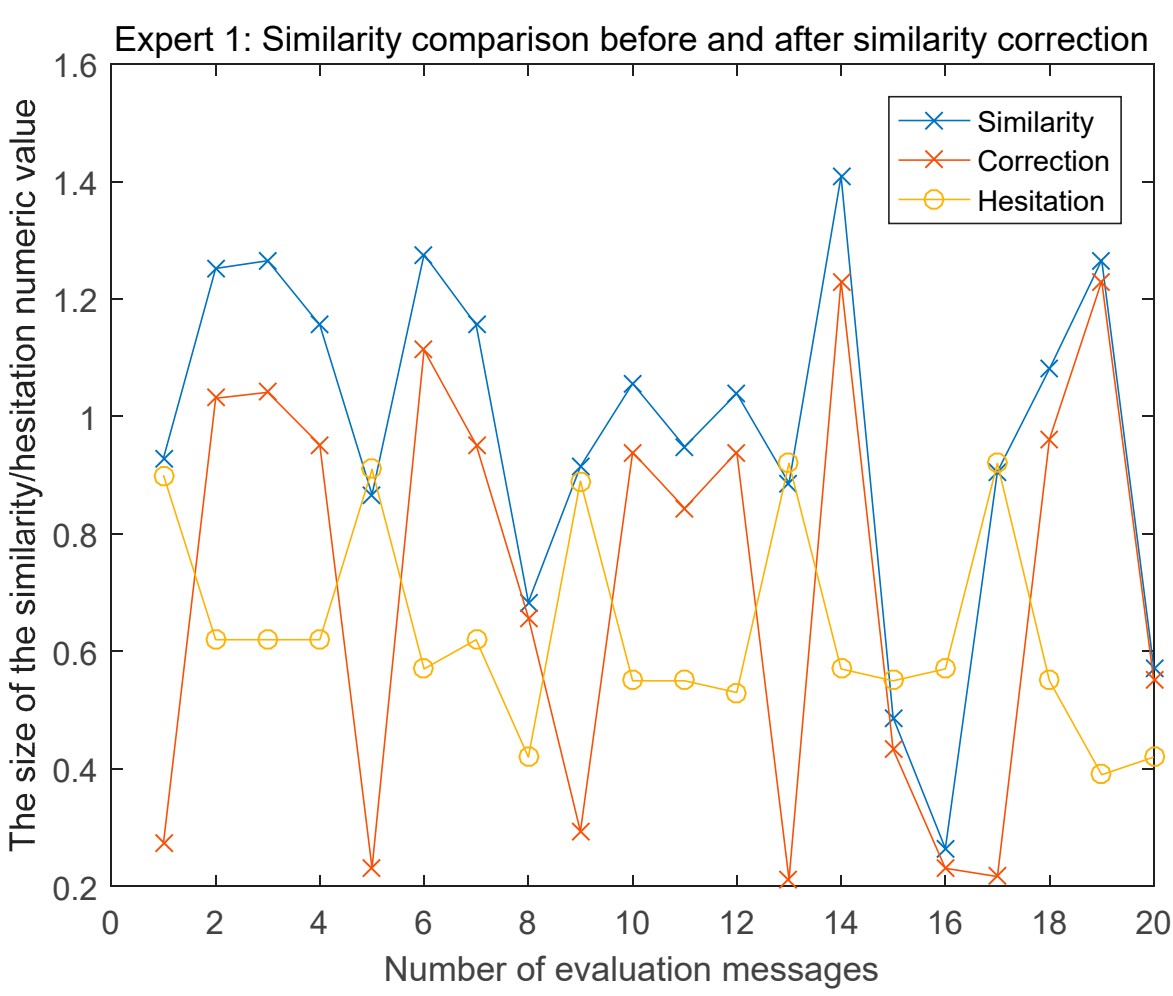

**Figure 1.** Before and After Similarity Correction of Expert 1.

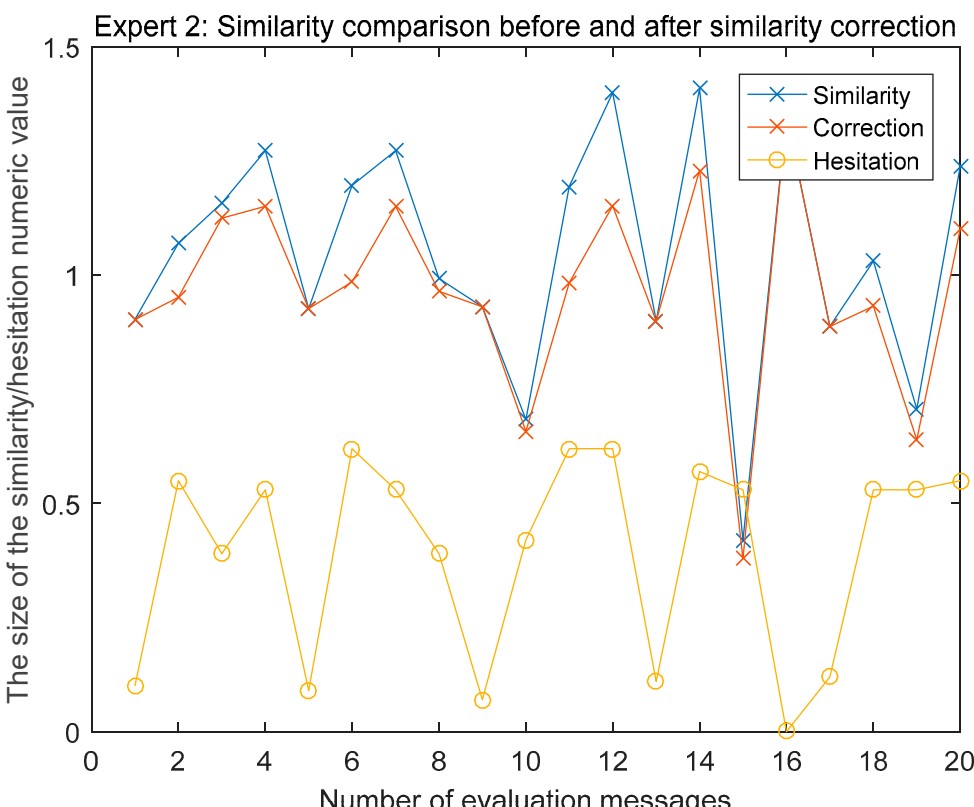

**Figure 2.** Before and After Similarity Correction of Expert 2.

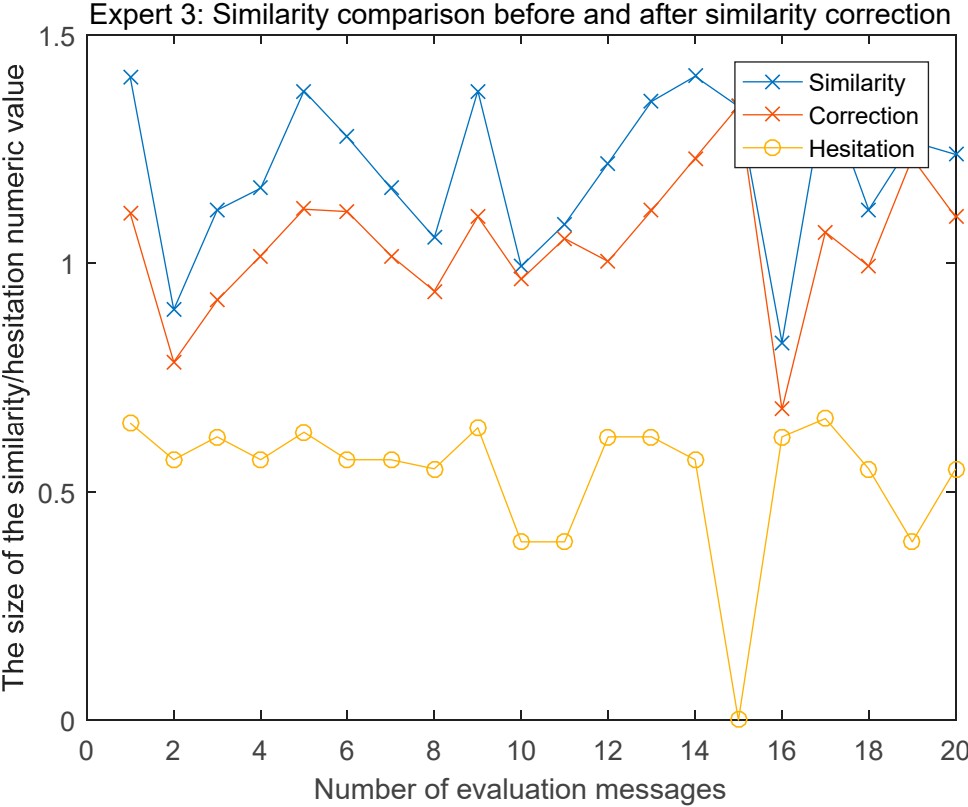

**Figure 3.** Before and After Similarity Correction of Expert 3.

Step 5: Determine the proximity matrix of an individual to other individuals. The proximity matrices $P^{(1)}$, $P^{(2)}$, and $P^{(3)}$ of $\varepsilon_1$, $\varepsilon_2$, and $\varepsilon_3$, respectively, are calculated using Equation (8) as Tables 11–13:

**Table 11.** Proximity Matrix $P^{(1)}$.

|  | **C1** | **C2** | **C3** | **C4** |
|---|---|---|---|---|
| $A_1$ | 0.6918 | 1.1860 | 1.2020 | 0.9709 |
| $A_2$ | 0.6383 | 1.2140 | 0.9709 | 0.5114 |
| $A_3$ | 0.6783 | 0.9338 | 0.8172 | 0.9288 |
| $A_4$ | 0.6281 | 1.4090 | 0.4450 | 0.2264 |
| $A_5$ | 0.6454 | 0.7629 | 1.1580 | 0.4292 |

**Table 12.** Proximity Matrix $P^{(2)}$.

|  | **C1** | **C2** | **C3** | **C4** |
|---|---|---|---|---|
| $A_1$ | 0.6760 | 0.7682 | 0.9417 | 1.2140 |
| $A_2$ | 0.6960 | 1.0580 | 1.2140 | 0.4937 |
| $A_3$ | 0.6770 | 0.5114 | 1.1090 | 1.3940 |
| $A_4$ | 0.6837 | 1.4090 | 0.3784 | 0.5517 |
| $A_5$ | 0.6368 | 0.9151 | 0.5700 | 1.1190 |

**Table 13.** Proximity Matrix $P^{(3)}$.

|  | **C1** | **C2** | **C3** | **C4** |
|---|---|---|---|---|
| $A_1$ | 1.4040 | 0.7539 | 1.0130 | 1.0640 |
| $A_2$ | 1.3570 | 1.2140 | 1.0640 | 0.9338 |
| $A_3$ | 1.3590 | 0.4937 | 0.7691 | 1.0760 |
| $A_4$ | 1.3230 | 1.4090 | 0.3987 | 0.7829 |
| $A_5$ | 1.3610 | 1.0190 | 1.1580 | 1.1190 |

Step 6: Determine the weights of experts. Using Equation (10), we derive the normalized weight matrix $\delta$ of experts $\varepsilon_1$, $\varepsilon_2$, and $\varepsilon_3$, as Table 14:

**Table 14.** Normalized Weight Matrix for Experts $\delta$.

|  | **C1** | **C2** | **C3** | **C4** |
|---|---|---|---|---|
| $\varepsilon_1$ | 0.1087 | 0.3489 | 0.3138 | 0.2415 |
| $\varepsilon_2$ | 0.4030 | 0.3147 | 0.2984 | 0.4147 |
| $\varepsilon_3$ | 0.4883 | 0.3364 | 0.3878 | 0.3437 |

### 5.2. Analysis of Expert Weights

In this section, we analyze the effectiveness of the proposed weight determination method from various perspectives, including similarity, similarity-corrected expert weight, and that in the literature. Through this analysis, we aim to evaluate the performance and reliability of the proposed method in determining the weights of experts.

### 5.2.1. Similarity Analysis

In order to facilitate data analysis, we extracted the evaluation information about experts $\varepsilon_1$ and $\varepsilon_2$ on alternative attribute C1, as presented in Table 15.

**Table 15.** Comparison of C1 evaluation information about experts 1 and 2.

| $\varepsilon_1$: **C1** | $\varepsilon_2$: **C2** |
|---|---|
| 0.3162, 0.3000, 0.9000 | 0.7533, 0.6500, 0.1000 |
| 0.2753, 0.3100, 0.9100 | 0.7794, 0.6200, 0.0900 |
| 0.3434, 0.3000, 0.8900 | 0.7893, 0.6100, 0.0700 |
| 0.2636, 0.2900, 0.9200 | 0.7432, 0.6600, 0.1100 |
| 0.2742, 0.2800, 0.9200 | 0.7589, 0.6400, 0.1200 |

It is seen from Table 15 that there is a clear difference between the degrees of hesitation of the two experts.

From the comparison of the distance between the evaluation information of the candidate attribute C1 and the ideal solution (mean) of experts $\varepsilon_1$ and $\varepsilon_2$ as presented in Table 16, we can find some existing problems, and the distances between the two are similar. If the traditional similarity-based weight assignment method is followed, then similar weights will be assigned to the two experts, $\varepsilon_1$ and $\varepsilon_2$. However, the evaluation of alternative attribute C1 by the two experts will have a large hesitation difference, which is obviously unreasonable. Therefore, we need to recalculate the weights by using similarity correction.

**Table 16.** Comparison of C1 Distances for Experts 1 and 2 with Mean Matrix.

| $\varepsilon_1$:d | $\varepsilon_2$:d |
|---|---|
| 0.4852 | 0.5112 |
| 0.5470 | 0.4873 |
| 0.4995 | 0.4833 |
| 0.5279 | 0.5141 |
| 0.5081 | 0.5262 |

The similarities of attribute C1 of experts 1 and 2 before corrections are presented in Table 17.

**Table 17.** Corrected C1 Similarities for Experts 1 and 2.

| $\varepsilon_1$:s | $\varepsilon_2$:s |
|---|---|
| 0.9290 | 0.9030 |
| 0.8672 | 0.9269 |
| 0.9147 | 0.9309 |
| 0.8864 | 0.9001 |
| 0.9061 | 0.8880 |

The similarities of experts $\varepsilon_1$ and $\varepsilon_2$ to variant attribute C1 after the excerpt are presented in Table 18.

From Table 18, it can be found that expert $\varepsilon_1$ has a large degree of hesitation towards alternative attribute C1, and their uncertainty is high, thus resulting in a higher correction effect. However, expert $\varepsilon_2$ has less hesitation towards alternative attribute C1, and their uncertainty is low, thus resulting in a smaller correction effect. It is also verified that the proposed method can effectively avoid the misjudgment situation faced by the traditional similarity-based method when the distance is similar.

**Table 18.** C1 Similarities of Experts 1 and 2 After Correction.

| $\varepsilon_1{:}S^*$ | $\varepsilon_2{:}S^*$ |
|---|---|
| 0.2732 | 0.9029 |
| 0.2314 | 0.9268 |
| 0.2935 | 0.9308 |
| 0.2118 | 0.9000 |
| 0.2165 | 0.8878 |

5.2.2. Analysis of Expert Weight with Similarity Correction

The weights of experts $\varepsilon_1$ and $\varepsilon_2$ on alternative attribute C1 are given in Table 19.

**Table 19.** C1 Weights of Experts 1 and 2.

| | **C1** |
|---|---|
| $\varepsilon_1$ | 0.1087 |
| $\varepsilon_2$ | 0.4030 |

Based on the similarity analysis, if we do not correct the similarity of the two experts for the attribute C1, we may assign similar weights to them based on their proximity to the ideal solution, assuming they have similar levels of expertise in that attribute. However, this approach overlooks the differences in hesitation within their evaluation information, leading to inaccurate weight assignment. To address this issue, we incorporate the corresponding hesitation in their evaluation information to correct the similarity and recalculate the weights. After making amendment, Table 19 shows that the weights of experts $\varepsilon_1$ and $\varepsilon_2$ on alternative attribute C1 are 0.1087 and 0.403, respectively, which better reflect the degree of uncertainty and expertise that exist in their evaluation information.

5.2.3. Comparative Analysis of Literature

In order to verify the effectiveness of the proposed method, we compare and analyze the weight assignment method presented in [14,23]. The obtained specific results are as follows: $\delta$ = [0.3252, 0.3754, 0.2544] in [23], $\delta$ = [0.2991, 0.3778, 0.3231] in [14], and the same in this paper is $\delta = \begin{bmatrix} 0.1087, & 0.3489, & 0.3138, & 0.2415 \\ 0.4030, & 0.3147, & 0.2984, & 0.4147 \\ 0.4883, & 0.3364, & 0.3878, & 0.3437 \end{bmatrix}$.

Among them, the subjective weight assignment method was used in [23], which does not fully consider the decision-making information in the expert evaluation matrix, making the use of decision-making information insufficient. In contrast, both the method used in [14] and that proposed in this paper are objective weight assignment methods, which are more objective and scientific in the concept of weight assignment. The method in [14] considers hesitation in the expert evaluation matrix, but its correction effect on the weight assignment is linear, while the same in this paper is nonlinear, which is more in line with the physical expression of hesitant attitudes of people in real life. Furthermore, in the decision-making process, each expert may be specialized in some specific attributes only. So, each expert should be given different weight values for different attributes [25]. In [14], the weights of different experts under different attributes were not distinguished, but they are distinguished in the method proposed in this paper. Proximity is not considered in the method presented in [14], as the number of experts involved in the group decision-making is limited. When some individual evaluations deviate from the normal range, it interferes with group decision-making. Therefore, in order to allocate the weights more reasonably, it is necessary to incorporate the proximity index (proximity) of individual experts to other experts during the weight calculation. In conclusion, building upon previous research, this paper extends the expert weight determination method by incorporating similarity and

proximity based on Pythagorean fuzzy sets, while also introducing hesitation correction into the similarity formula. The proposed methodology not only addresses the limitations of previous similarity-based approaches but also ensures that the expert weights are more objective and reasonable. As a result, the method proposed in this paper is superior in determining the weights of experts and aligns better with the practical needs of decision-making processes.

*5.3. Group Decision-Making*

Step 1: Determine the aggregate PFDM, $\mathcal{X} = (x_{ij})_{l*n}$. The expert weights $\delta_j^{(k)}$ and Equation (2) obtain $\mathcal{X} = (x_{ij})_{l*n}$ as Table 20:

**Table 20.** Aggregated Pythagorean Fuzzy Decision Matrix $\mathcal{X}^{(k)}$.

|  | C1 | C2 | C3 | C4 |
|---|---|---|---|---|
| $A_1$ | 0.6658, 0.4939, 0.5592 | 0.5943, 0.5018, 0.6285 | 0.7650, 0.3401, 0.5469 | 0.4200, 0.6953, 0.5832 |
| $A_2$ | 0.6993, 0.4642, 0.5436 | 0.3343, 0.7308, 0.5952 | 0.4333, 0.6771, 0.5948 | 0.7324, 0.4424, 0.5175 |
| $A_3$ | 0.6873, 0.4920, 0.5344 | 0.6878, 0.4971, 0.5290 | 0.7562, 0.3895, 0.5258 | 0.6065, 0.4879, 0.6278 |
| $A_4$ | 0.6684, 0.5006, 0.5501 | 0.2000, 0.8000, 0.5657 | 0.9836, 0.0231, 0.1788 | 0.9884, 0.0145, 0.1509 |
| $A_5$ | 0.6547, 0.5031, 0.5641 | 0.6211, 0.4995, 0.6039 | 0.8390, 0.2967, 0.4561 | 0.7351, 0.3407, 0.5862 |

Step 2: Determine the aggregation-weighted PFDM, $\mathcal{X}' = (x'_{ij})_{l\times m}$ as Table 21. According to Table 20 and Equation (12), $\mathcal{X}' = (x'_{ij})_{l\times m}$ can be calculated, wherein the attribute weight matrix $\mathcal{W}$ is given as follows:

$$\mathcal{W}_{\{C_1,C_2,C_3,C_4\}} = \begin{bmatrix} (0.8412, & 0.3457, & 0.4158) \\ (0.7871, & 0.3545, & 0.5048) \\ (0.7871, & 0.3545, & 0.5048) \\ (0.7563, & 0.3800, & 0.5326) \end{bmatrix}^T$$

**Table 21.** Aggregation-Weighted Pythagorean Fuzzy Decision Matrix.

|  | C1 | C2 | C3 | C4 |
|---|---|---|---|---|
| $A_1$ | 0.5601, 0.5782, 0.5933 | 0.4677, 0.5881, 0.6598 | 0.6021, 0.4762, 0.6408 | 0.3177, 0.7470, 0.5840 |
| $A_2$ | 0.5882, 0.5561, 0.5871 | 0.2631, 0.7698, 0.5815 | 0.3410, 0.7256, 0.5976 | 0.5539, 0.5585, 0.6175 |
| $A_3$ | 0.5781, 0.5768, 0.5772 | 0.5414, 0.5846, 0.6043 | 0.5952, 0.5082, 0.6224 | 0.4587, 0.5899, 0.6645 |
| $A_4$ | 0.5622, 0.5832, 0.5863 | 0.1574, 0.8278, 0.5385 | 0.7742, 0.3552, 0.5239 | 0.7476, 0.3802, 0.5446 |
| $A_5$ | 0.5507, 0.5851, 0.5952 | 0.4889, 0.5863, 0.6459 | 0.6604, 0.4502, 0.6010 | 0.5559, 0.4937, 0.6688 |

Step 3: Determine the positive and negative ideal solutions. In this case, the revenue type attribute is $J_1 = \{C_1, C_2, C_4\}$ and the cost type attribute is $J_2 = \{C_3\}$. According to the Equations (14)–(17), the positive ideal solution $\mathcal{A}^+$ and negative ideal solution $\mathcal{A}^-$ can be obtained as follows:

$$\mathcal{A}^+ = \{(0.5582,0.5561,0.5871), (0.5414,0.5846,0.6043), \\ (0.3410,0.7256,0.5976), (0.7476,0.3802,0.5466)\}.$$

$$\mathcal{A}^- = \{(0.5507,0.5851,0.5952), (0.1574,0.8278,0.5385), \\ (0.7742,0.3552,0.5239), (0.3177,0.7470,0.5840)\}.$$

Step 4: Determine the distance between the positive and negative ideal solutions and the alternatives. Using Equations (18) and (19), we can obtain the following from $\mathcal{A}^+$ and $\mathcal{A}^-$:

$$D(\mathcal{A}_i, \mathcal{A}^+) = \{0.2618, 0.1630, 0.1982, 0.2734, 0.1971\},$$

$$D(\mathcal{A}_i, \mathcal{A}^-) = \{0.1799, 0.2552, 0.2103, 0.2188, 0.2157\}.$$

Step 5: According to Equation (20), the composite evaluation index of alternatives $\rho(\mathcal{A}_i)$ can be obtained as follows:

$$\rho(\mathcal{A}_i) = \{-0.9014, 0, -0.3922, -0.8206, -0.3645\}.$$

The optimal solution is $\mathcal{A}_2 \succ \mathcal{A}_5 \succ \mathcal{A}_3 \succ \mathcal{A}_4 \succ \mathcal{A}_1$. Therefore, the alternative in this data is the most suitable location for a university campus in Fezrabad.

### 5.4. Verification of Theoretical Carrying Capacity

The following tests the adaptability of the ideas in this article through extreme conditions: add an expert $\varepsilon_4$ who does not know much about the field and has a consistently high level of hesitation in their judgments.

Evaluation matrix for $\varepsilon_4$ in Table 22:

**Table 22.** Pythagorean Fuzzy Decision Matrix $\mathcal{X}^{(4)}$.

|  | C1 | C2 | C3 | C4 |
|---|---|---|---|---|
| $A_1$ | 0.0995, 0.1000, 0.9900 | 0.2114, 0.1200, 0.9700 | 0.4145, 0.1900, 0.8900 | 0.0995, 0.1000, 0.9900 |
| $A_2$ | 0.1720, 0.1000, 0.9800 | 0.0883, 0.1100, 0.9900 | 0.4093, 0.1500, 0.9000 | 0.4231, 0.1700, 0.8900 |
| $A_3$ | 0.3903, 0.1400, 0.9100 | 0.4055, 0.1600, 0.9000 | 0.2480, 0.1300, 0.9600 | 0.3356, 0.1500, 0.9300 |
| $A_4$ | 0.2681, 0.1600, 0.9500 | 0.2883, 0.1200, 0.9500 | 0.0995, 0.1000, 0.9900 | 0.2883, 0.1200, 0.9500 |
| $A_5$ | 0.2530, 0.1200, 0.9600 | 0.3438, 0.1300, 0.9300 | 0.3474, 0.1200, 0.9300 | 0.3969, 0.1200, 0.9100 |

Weight matrix after adding $\varepsilon_4$:

Sort the results after the addition of $\varepsilon_4$ as $\mathcal{A}_2 \succ \mathcal{A}_5 \succ \mathcal{A}_3 \succ \mathcal{A}_4 \succ \mathcal{A}_1$. Since expert $\varepsilon_4$ does not know much about the field, his/her hesitation in the evaluation of information is high. According to the concept of this article, expert $\varepsilon_4$ is given a smaller weight so as to not affect the sorted final result. It can be seen from the results presented in Table 23 that the weight of $\varepsilon_4$ is given a small value due to its overall hesitation, and its evaluation information does not adversely affect the original ranking results, which fully verifies the validity of the weight assignment and the rationality of the decision-making process of the proposed method.

**Table 23.** Normalized Weight Matrix for Experts $\delta$.

|  | C1 | C2 | C3 | C4 |
|---|---|---|---|---|
| $\varepsilon_1$ | 0.1394 | 0.3283 | 0.3056 | 0.2250 |
| $\varepsilon_2$ | 0.3412 | 0.3017 | 0.2714 | 0.3889 |
| $\varepsilon_3$ | 0.4582 | 0.3075 | 0.3624 | 0.3209 |
| $\varepsilon_4$ | 0.0612 | 0.0624 | 0.0605 | 0.0651 |

### 5.5. Validation of Methodology Using Real Expert Data from Reference [23]

In this section, we validate the effectiveness and applicability of our proposed method by incorporating real expert data. The inclusion of real expert data allows us to further assess the performance and reliability of our approach in real-world scenarios.

We obtained the real expert data from the referenced source (Literature [23]), which provides a comprehensive dataset collected from domain experts in the field. The data consist of expert evaluations, ratings, or opinions related to the decision-making problem addressed in this study.

The National Academy of Science of Pakistan is dedicated to the national youth education and is the largest educational network in Pakistan. In order to meet the growing educational needs in Faisalabad, they decided to establish a university campus there to provide quality education and a good learning environment for students. After a visit to the city and a pre-assessment, they planned to choose one of five alternative sites $\mathcal{A}_i (i = 1, 2, \ldots, 5)$ as the best place to build a university campus. In order to address this decision-making issue, the owners of the educational institutions formed a committee of three experts, including a legal expert, $\varepsilon_1$, an investment expert, $\varepsilon_2$ and a population expert, $\varepsilon_3$, to evaluate the alternative locations based on the following four attributes:

C1: Policy and theoretical perspective;
C2: Convenience and livability of teachers and students;
C3: Construction cost;
C4: Economic development of the region.

Pedro Ernesto Hospital in Rio de Janeiro, Brazil is a well-reputed hospital, focused and dedicated to providing healthcare facilities in the region. It consists of number of surgical and clinical departments and is considered a reference center for computer tomography, nuclear medicine, and hemodialysis. The hospital is planning to purchase healthcare technology. After a pre-analysis of the resource availability and financial condition of the hospital, five technologies, $\mathcal{A}_1$ (magnetic resonance image), $\mathcal{A}_2$ (single-positron emission computer tomography), $\mathcal{A}_3$ (video laparoscope), $\mathcal{A}_4$ (mamograph), and $\mathcal{A}_5$ (cardio-angiograph), are identified as feasible alternatives for purchasing, and all are recognized as well-known procedures. The committee of professional experts consisted of two medical doctors, $\varepsilon_1$ and $\varepsilon_2$, and an equipment acquisition expert, $\varepsilon_3$, who evaluated the feasible alternatives with respect to the following four criteria:

C1: Benefit population;
C2: Dependence on maintenance;
C3: Professional and community demand;
C4: Important for improving patients' health;
C5: Expected advantages in health outcomes.

The ranking results obtained using the methodology proposed in this paper and the methodology from reference [23] are presented in Tables 24 and 25. From Table 25, it can be observed that the ranking results obtained using the proposed methodology are consistent with those obtained using the methodology from reference [23]. The similarity in the ranking results indicates the rationality and effectiveness of our proposed method. It also demonstrates the applicability of our methodology across different datasets and confirms the reliability of the ranking results obtained through our approach.

**Table 24.** Comparing Ranking Results with Reference [23].

| Method | Ranking Results |
|---|---|
| Reference [23] | $A_3 \succ A_2 \succ A_5 \succ A_1 \succ A_4$ |
| In this paper | $A_3 \succ A_2 \succ A_1 \succ A_4 \succ A_5$ |

**Table 25.** Comparing Ranking Results with Reference [23].

| Method | Ranking Results |
|---|---|
| Reference [23] | $A_3 \succ A_4 \succ A_5 \succ A_1 \succ A_2$ |
| In this paper | $A_3 \succ A_4 \succ A_5 \succ A_1 \succ A_2$ |

Furthermore, based on the analysis of Table 24, it can be observed that due to the objective determination of decision-makers' weights in our method compared to the subjective determination used in the prior methodology from reference [23], it is normal to observe slight differences in the ranking results, particularly in cases A1, A4, and A5. However, these differences serve as indirect validation of the effectiveness of our proposed method. The slight variations highlight the robustness and adaptability of our methodology in handling different decision-making scenarios. Consequently, our weighting method, based on the improved hesitation of Pythagorean fuzzy sets, is considered more rational and effective, building upon the demonstrated effectiveness of our method in these cases.

In conclusion, the findings support the validity and applicability of our methodology and highlight its advantages over existing approaches, as discussed above. The consistent ranking results between our method and the methodology from reference [23], along with the slightly divergent results in specific cases, reinforce the strength of our objective approach to determine decision-makers' weights. The proposed weighting method, based on the improved hesitation of Pythagorean fuzzy sets, offers a more rational and effective solution for decision-making processes.

## 6. Summary

Aiming at the MCGDM problem of multi-attribute group decision-making with unknown expert weights and attribute evaluation information being PFN, a method for determining the weight of experts after comprehensively considering the similarity and proximity of the evaluation information after correction is proposed. The results of expert weight analysis and theoretical carrying capacity verification, as well as the validation of our methodology using real expert data from reference [23], collectively demonstrate the effectiveness and reliability of our proposed approach:

- The proposed method takes into account the similarity degree and the corrected similarity, which effectively reflects the difference between the degree of uncertainty and professionalism of experts in the evaluation information.
- The inclusion of similarity and corrected similarity improves the rationality and accuracy of the final decision-making process.

However, this article also has some limitations. Firstly, the proposed method still relies on decision-making information, which may lead to inaccuracies in determining expert weights when the decision-making information is not reliable. Secondly, the dataset used in this experiment is small, and further investigation is needed to validate the applicability of the method on larger datasets in real-world scenarios. In conclusion, the weight determination method based on the improved hesitation of Pythagorean fuzzy sets presented in this paper offers a novel solution to the multi-attribute group decision-making problem, demonstrating high accuracy and practicality. However, there is still room for future research and expansion in this field [27,28].

**Author Contributions:** Conceptualization, X.D. and K.L.; methodology, X.D., R.Z. and Y.L.; software, K.L. and S.Q.; validation, X.D. and K.L.; formal analysis, K.L., R.Z. and Y.L.; investigation, K.L. and S.Q.; resources, X.D. and S.Q.; data curation, X.D., R.Z. and K.L.; writing—original draft preparation, K.L., R.Z. and Y.L.; writing—review and editing, X.D. and K.L.; visualization, K.L., R.Z. and S.Q.; supervision, S.Q. and Y.L.; project administration, X.D. and K.L.; funding acquisition, X.D. All authors have read and agreed to the published version of the manuscript.

**Funding:** The project is sponsored by "Liaoning BaiQianWan Talents Program", grant number 2018921080.

**Data Availability Statement:** The processed data required to reproduce these findings cannot be shared as the data also form part of an ongoing study.

**Conflicts of Interest:** The authors declare no conflict of interest.

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
