# Peer review of "A Weighting Method Based on the Improved Hesitation of Pythagorean Fuzzy Sets"

_electronics, doi:10.3390/electronics12133001_

Round 1
Reviewer 1 Report
1- make the abstract shorter.
2- pls rewrite from lines 100 to line 141, use articles while explaining.
3- pls. stick with the four significant digits in formating numbers in all tables
4- you need to expand the writing for this subsection
5.1 Determination of expert weights
5- figure 2 The caption is not clear
6- please check tables with symbols at the end that looks above the writing and not at the same level for example the title of table 9.
need to review the English deeply.
Reviewer 2 Report
The paper addresses a problem in multi-attribute group decision-making (MCGDM) where expert weights are unknown and attributes are evaluated based on Pythagorean fuzzy numbers. The authors argue that existing expert weight determination methods do not make sufficient use of the hesitation involved with the decision information, which may lead to biased weight assignment. To address this issue, they propose a weight determination method that improves the hesitation of the Pythagorean fuzzy set. Here are my comments:
1. The paper should be proofread first. The content has many typo and symbol errors and many symbols are not inline and misused. For example, in page 4, the authors used { } as the matrix representation, however, {} should be the set representation, and [] should be the matrix representation; page 5, the authors used П to denote pi instead of π.
2. The structure of Introduction should be modified. Ignore the math description and focus on the contribution and novelty of the paper.
3. In page 3, the last line … equation (1), equation (2) should be equations (2) and (3).
4. In page 5, Eq. (4) should be Eq. (8). Sometimes you used equation and sometimes you used Eq.
5. Inconsistency of numerical decimals, e.g., Tables 1-4.
6. I don’t think the author used the appropriate case in this paper. There are only three experts but the results show significant hesitation difference or inconsistency. The easy way is to understand the reasons for the difference or use Delphi to make the results consistent instead of complicating the problem using similarity correction. In addition, the result also suggests that these men are not real experts. This can be verified on page 17; said expert 4 does not know much about the field. You should pick a real expert rather than adjust the results.
In conclusion, the authors should rewrite the paper to be considered in the journal.
Proofread is needed.
Reviewer 3 Report
Dear authors:
a) please include with bullet points in the introduction the novelities of the paper. In bullet points, in short form
b) Same in the end - in the conclusion section
c) Please double check again for some minor english grammar corrections. Be careful with commas!
d) in the conclusion, please expand your literature review. I would recommend including as future work the following two papers.
paper1: The paper below shows a methodology which would be interesting to apply to an extension of the paper
Giannelos, S.; Borozan, S.; Strbac, G. A Backwards Induction Framework for Quantifying the Option Value of Smart Charging of Electric Vehicles and the Risk of Stranded Assets under Uncertainty. Energies 2022, 15, 3334. https://doi.org/10.3390/en15093334
paper2: similarly it would be interesting to see this applied too:
J. M. Mendel, H. Hagras, H. Bustince and F. Herrera, "Comments on “Interval Type-2 Fuzzy Sets are Generalization of Interval-Valued Fuzzy Sets: Towards a Wide View on Their Relationship”," in IEEE Transactions on Fuzzy Systems, vol. 24, no. 1, pp. 249-250, Feb. 2016, doi: 10.1109/TFUZZ.2015.2446508.
Some additional comments:
1.Can you place some extra references in the 3rd section ?
2. The figures need improvement. Try to increase readability by improving the captions.
Reviewer 4 Report
The proposed manuscript is focused on the problem of multi-attribute group decision-making based on the improved hesitation of Pythagorean fuzzy sets use. The problem is interesting and adheres to the scientific scope of the Journal.
The authors clearly presented the main research gap and contribution of this text. The structure of the paper is proper and clearly presents the new methodology approach confirmed by the case studies.
The mathematical formulations are clearly presented. They are consistent with a theoretical approach.
The results of the case studies are properly described and supported by the graphs and tables.
The article can be accepted in the present form.
Round 2
Reviewer 2 Report
The authors have almost replied to all my comments but I still ask the authors to consider if they move the mathematics to section 2 to describe the question and problem of the previous method.
Minor editing of English language required.
